Why the long face? Comparative shape analysis of miniature, pony, and other horse skulls reveals changes in ontogenetic growth

Heck Laura 1
Sanchez-Villagra Marcelo R. 1
Stange Madlen stange.madlen@gmail.com 2
1 Palaeontologisches Institut und Museum, University of Zürich , Zürich , Switzerland
2 Department of Biology & Redpath Museum, McGill University , Montréal , Quebec , Canada
Wedel Mathew
Electronic publication date: 2019 Sep 16
Publication date: 2019
Volume: 7
Electronic Location ID: e7678
Received 2019 Apr 12; Accepted 2019 Aug 15
Copyright: ©2019 Heck et al.
Copyright year: 2019
Copyright holder: Heck et al.
License: This is an open access article distributed under the terms of the Creative Commons Attribution License, which permits unrestricted use, distribution, reproduction and adaptation in any medium and for any purpose provided that it is properly attributed. For attribution, the original author(s), title, publication source (PeerJ) and either DOI or URL of the article must be cited.
License URL: https://creativecommons.org/licenses/by/4.0/

Keywords: Domestication, Geometric morphometrics, Allometry, Pony, Falabella, Cranium, Equus, Ontogeny

Funding: Swiss National Science Foundation 31003A_169395 Swiss National Science Foundation P2ZHP3_178108 SYNTHESYS AT-TAF-5786 ‘Morphological disparity and ontogenetic allometry in domesticated horses Marcelo R. Sánchez-Villagra, Laura Heck, and Madlen Stange were supported by Swiss National Science Foundation grant no. 31003A_169395 granted to Marcelo R. Sánchez-Villagra. Madlen Stange further received funding from Swiss National Science Foundation grant no. P2ZHP3_178108. Support for the collection visit at the NHW Vienna, Austria was granted by SYNTHESYS funding AT-TAF-5786 “Morphological disparity and ontogenetic allometry in domesticated horses” to Laura Heck. The funders had no role in study design, data collection and analysis, decision to publish, or preparation of the manuscript.

==============================
Background

Much of the shape variation found in animals is based on allometry and heterochrony. Horses represent an excellent model to investigate patterns of size-shape variation among breeds that were intentionally bred for extreme small and large sizes.

Methods

We tested whether ponies (wither height < 148 cm) have a diverging size-shape relationship in skull shape as compared to regular-sized horse breeds (wither height > 148 cm, here-after called horses) during ontogenetic growth. We used a dataset of 194 specimens from 25 horse and 13 pony breeds, two of which are miniature breeds (wither height < 96.5 cm)—Falabella, Shetland. We applied three-dimensional geometric morphometrics, linear measurements, and multivariate analyses (Procrustes ANOVAs) to quantitatively examine and compare the ontogenetic trajectories between pony and horse breeds with an emphasis on the miniature breeds as an extreme case of artificial selection on size. Additionally, we tested for juvenile characteristics in adult horse and miniature breeds that could resemble “paedomorphosis”—retention of juvenile characteristics in adult stage; e.g. large eyes, large braincase-to-face-relationship, and large head-to-body relationship.

Results

Allometric regression of size on shape revealed that 42% of shape variation could be explained by variation in size in all breeds. The ontogenetic trajectories of ponies and horses vary in slope and therefore in rate of change per unit size, and length. The differences in trajectory lengths and slopes result in ponies having a similar skull shape in an older age stage than horses of the same size in a younger age stage. This pattern could cause the generally perceived “paedomorphic” appearance of ponies. Miniature breeds have larger heads in relation to wither height compared to horses, a non-paedomorphic feature in horses specifically. Also, rostra (faces) are longer in adult individuals than in juveniles across all kinds of breeds. This pattern can be explained by the long-face hypothesis for grazing ungulates and could possibly be caused by the mismatch of selection by humans for shorter rostra and the dentition of ruminants.

Conclusions

Miniature breed specimens do not exhibit any of the classical mammalian “paedomorphic” features (large orbits, large heads), except for the adult Falabella that has enlarged orbits, possibly because they are herbivorous ungulates that are affected by functional and metabolic constraints related to low nutrient-food consumption. Instead ponies, including miniature breeds, have faster and shorter ontogenetic growth compared to horses, resulting in adult pony skulls looking in part like juvenile horse skulls.

Introduction

Allometry, shape change associated with size change, accounts for much of newly generated shape changes (Wilson, 2018). Patterns of shape variation and allometry during ontogeny, that is the size-shape relationship during an organisms growth, have been studied in many domesticated animals such as dogs (Wayne, 1986; Morey, 1992; Goodwin, Bradshaw & Wickens, 1997; Geiger et al., 2017; Werneburg & Geiger, 2017), pigs (Hilzheimer, 1926; Evin et al., 2017), sheep (Geist, 1974), guinea pigs (Kruska & Steffen, 2013), and horses (Radinsky, 1984; Goodwin, Levine & McGreevy, 2008). Recent advances in the study of shape variation, especially through the advancement of statistical methods for the analysis of multivariate geometric morphometric data, have increased our knowledge on how shape variation arises and how variation is patterned (Adams et al., 2004; Adams, Rohlf & Slice, 2013).

Horses exhibit a large size range from 84 cm (Falabella) to 178 cm wither height (Shire). Ponies are defined by a wither height less than 148 cm (FEI, 2016). Ponies are derived from selection on larger breeds (Hendricks, 2007). Extremely small ponies, so-called miniature breeds, formally belong to the group of ponies. Miniature breeds are defined as horse breeds with a wither height of less than 96.52 cm (38 in) (World Class Miniature Horse Registry, 2018) and have been purposely bred for extreme small size. The smallest horse breed, the Falabella, originates from Argentina. The breed was first mentioned in the middle of the 19th century when very small individuals of Criollo horses were encountered in the Argentinian Pampa (Hendricks, 2007). After obtaining a few individuals and starting a breeding program with miniature horses, the name giver of the breed, Juan Falabella, added small individuals of English Thoroughbred, Criollo, and the also miniature breed Shetland pony to achieve a harmonious conformation with a wither height lower than 84 cm. The Shetland pony, which was strongly interbred with the Falabella due to its small wither height (max. 106 cm), originates from the Shetland Islands, Scotland. It is among the oldest known horse breeds and was mostly bred locally on the islands for croft works. When an act of British Parliament, however, prohibited child labour in the coalmines in 1847, the demand for these small robust ponies, as a replacement, increased drastically. Over the last century, numerous individuals have been exported, mainly being used for driving or as a first mount for children (Hendricks, 2007).

In the following, we investigate ontogenetic trajectories of ponies (wither height <148 cm) including two miniature breeds, and regular-sized horse breeds (wither height >148 cm) using three-dimensional geometric morphometrics (3D GM). We will refer to regular-sized horse breeds as ‘horses’ for simplicity. If not stated otherwise ponies always include the miniature breeds. We will highlight two miniature breeds—Falabella and Shetland, and their position in morphospace. Osteological samples of the Falabella are so rare that we have quantified skull shape from the only complete skull specimen available in public museum collections. The comparative analysis of these extreme cases of miniaturisation with other breeds will give us insights into differential growth patterns in horses due to artificial selection for size.

Further, we aim to shed light on the perceived juvenile appearance of miniature and pony breeds when compared to horses by assessing “paedomorphosis” in its broadest sense—the resemblance of an adult form to a juvenile of a sister group. We are aware that breeds are not individual species, for that reason we do not claim to study paedomorphosis in the strict sense. However, there are certain features in Falabella and Shetland adult skulls that raise the question, whether they have retained juvenile appearance. For example, the adult Falabella skull exhibits a round braincase that can partly be seen in the other miniature breed—the Shetland, but not in the larger Welsh (Fig. 1). Generally, juvenilized phenotypic features are differences in body proportions, e.g., a larger head and shorter limbs (Gould, 1980), and differences in cranial proportions, such as larger eyes, a more prominent and bulging cranium, and a short rostrum in combination with an enlarged braincase (Gould, 1980; Wayne, 1986; Tamagnini, Meloro & Cardini, 2017; Evin et al., 2017). In particular, the shortening of the face portion of the skull in smaller taxa versus the elongation of the face portion in larger related taxa has gained some attention over the past years and has been formalized as the cranial evolutionary allometry hypothesis (CREA) (Tamagnini, Meloro & Cardini, 2017). To approach the quantification of juvenilized shape we use linear measurements derived from the 3D GM dataset for all age classes and all breeds and calculate ratios of length that reflect the typical paedomorphic traits of larger eyes, shorter face, and smaller head to body ratio in ponies.

Figure 1 Cranial shape comparison among two miniature breeds and a pony through ontogeny.

Examples of different cranial shapes during ontogeny from lateral view if available for each age class (0–6) for (A) Falabella and (B) Shetland (miniature breeds); and (C) Welsh (pony); each stage is represented by a different individual and all crania are scaled to the same length for comparison. Photographs by Laura Heck.

Materials & Methods

Specimens analyzed and determination of age classes

A total of 194 juvenile and adult crania were analyzed (Table S1). We examined specimens from the following collections: Museum für Naturkunde Berlin (MfN Berlin, Germany), Institut für Haustierkunde (Christian-Albrechts-Universität of Kiel, Germany), Museum für Haustierkunde “Julius Kühn” (University of Halle, Germany), Naturhistorisches Museum Wien (NHW Vienna, Austria), and Museo de la Plata (MLP La Plata, Argentina). The dataset includes 38 horse breeds, ranging from the smallest (Falabella) to the largest breed (Shire) (Table S1). Of the 38 breeds, 13 are considered ponies two of which are miniature breeds as indicated in brackets: Bosnian pony (bos), Exmoor pony (exm), Falabella (fab) (miniature), German Riding pony (grp), Icelandic Horse (ice), Indian pony (ind), Konik (kon), Mongolian (mon), Scottish pony (scp), Shetland pony (she) (miniature), Togo pony (tog), and Welsh (wel). 25 breeds are considered regular-sized breeds or horses: Anglo-Norman (ano), Arab (arb), Birkenfelder (bif), Belgian Draft (blg), Clydesdale (cds), Galician Farm Horse (gbh), Grisons (Graubündner) (grb), Hannoverian (han), Hackney (hny), Holstein (hol), Hungarian (hun), Huzule (huz), Kladrubian (kdr), Kosarian (kos), Lipizzan (lpz), Nonius (nos), Norik (nor), Oldenburgian (odb), Pinzgau (piz), Polish Farm Horse (pll), Seneca Sarajevo (ses), Shire (shi), Styrian (stm), Suffolk (suf), English Thoroughbred (thb), Trakehner (trk).

Prior to analyses, each specimen was categorized into an age class from 0 to 6 using an identification key for dental eruption (Habermehl, 1975) with: 0—dental eruption after birth, 1—eruption of the first pair of deciduous incisors, 2—eruption of the second pair of deciduous incisors, 3—eruption of the third pair of deciduous incisors, 4—eruption of the first molar, 5—eruption of the second molar, and 6—eruption of the third molar (Table S1). Weaning occurs around the end of age class 2 and the beginning of age class 3 (at around six months), while sexual maturity is reached at the beginning of age class 4 (at around one year) and skeletal maturity is reached in age class 6 (around 4 years). Notably, the dataset contains only a single specimen of Falabella, to our knowledge the currently only complete skull from a museum collection to be measured.

Collection of shape data

Cranial shapes were analysed using landmark-based geometric morphometric (GMM) approaches. The crania were measured in three-dimensions (3D) using a MicroScribe® MLX6 (Revware, Inc., Raleigh, North Carolina, USA; accuracy: 0.076 mm) and a total of 60 type I and type II landmarks (Bookstein, 1990) (Table S2, Fig. S1) were collected. The dorsal and ventral sides of the crania were measured separately and the landmark datasets were subsequently combined using three reference landmarks (numbered 1, 2, and 33, Fig. S1) in the Microscribe® software MUS (Revware, Inc., Raleigh, North Carolina, USA). All subsequent analyses were conducted using R v.3.5.2 (R Core Team, 2018) and related R packages for the analyses of geometric morphometric data: geomorph v. 3.1.2 (Adams et al., 2018) and shapes v. 1.2.4 (Dryden, 2018). The code that was used for the following analyses of 3D GM data can be found as a supplement in Code S1.

3D Geometric morphometric analyses

General Procrustes Analysis (GPA) (Rohlf & Slice, 1990) was performed on the 3D shape data to eliminate the effects of size, orientation, and scaling. GPA translates, rotates, and scales all specimens’ coordinates so their centroids coincide and are scaled to unit centroid size, and the squared summed distances between matching landmarks are minimized. Due to its bilateral symmetry, only the symmetric component of the cranium was used in the subsequent analyses (Klingenberg, Barluenga & Meyer, 2002; Kolamunnage & Kent, 2003). We calculated centroid size for each landmark configuration.

To visualize morphospace occupation of the age classes of miniature breeds and horses along major axes of variance, we performed a principal component analysis (PCA) using the co-variance matrix of Procrustes scores retained from the GPA. We calculated mean shapes for each age class of horse breeds as well as for the Shetland and Falabella to visualize shape differences with age.

Characterizing cranial ontogenetic shape trajectories of ponies and horses

For subsequent analyses of ontogenetic size-shape co-variation (allometry) between and within pony and horses we performed linear regressions (Procrustes ANOVA) of shape (Procrustes coordinates) on logarithmized centroid size, and a grouping factor, type = H/P, which denotes the group affiliation of each breed to either horse (H) or pony (P) breed as indicated in Table S1 and raw coordinates file. To inspect the inter-specific allometric relationship between pony and horse breeds we used two methods, the regression of shape on size resulting in a plot of regression residuals on log centroid size (Drake & Klingenberg, 2008) and the predicted shape approach (Adams & Nistri, 2010) that plots the first principal component from a regression of predicted shape values on log centroid size. We applied a test for homogeneity of slopes (HOS) when the interaction term of log(size) and type was significant during Procrustes ANOVA (Collyer & Adams, 2013; Collyer, Sekora & Adams, 2015; Adams & Collyer, 2016). The HOS test allowed us to determine whether pony and horse breed ontogenetic allometries differed in slope distance (amount of shape changes with size), slope angles (direction of shape change), or intercept.

Testing features of paedomorphosis in miniature breeds using linear measurements

Additionally, to complement the multivariate statistical analyses for differences in ontogenetic trends between ponies and horses, we aimed to test for features of “paedomorphosis” in the miniature breeds using only linear measurements. We define “paedomorphosis” loosely here, as a general resemblance of adults in miniature breeds to juveniles in all other breeds. Typical phenotypic features of paedomorphosis are differences in body proportions, e.g., a larger head and shorter limbs (Gould, 1980), and differences in cranial proportions including larger eyes, a more prominent and bulging cranium, and a short rostrum in combination with an enlarged braincase (Gould, 1980; Wayne, 1986; Tamagnini, Meloro & Cardini, 2017; Evin et al., 2017). Paedomorphism has been claimed to describe some differences among horse breeds (Budiansky, 1997; Goodwin, Levine & McGreevy, 2008), however the long-face hypothesis of grazing ungulates (Spencer, 1995), that postulates that longer faces are observed in smaller forms of grazing ungulates, if also true for horses, could explain lack of signs of paedomorphism in the rostrum. We calculated interlandmark distances (specified below) from the three-dimensional dataset in the R package geomorph (Adams et al., 2018) and calculated the ratios for the following three traits:

Larger eyes

To test whether miniature breeds exhibit larger orbits (eyes) than regular-sized breeds relative to their respective cranial lengths, we calculated the ratio of orbit length to cranial length from measurements of the orbit diameter (LM 15–17, Table S2, Fig. S1) and total cranial length (LM 37–58, Table S2, Fig. S1).

Figure 2 Cranial mean shapes for adult Falabella and Shetland, and for each analyzed age class of horses.

Cranial shapes in (A) lateral and (B) dorsal view for the average shape of each age class (0–6) of all horse specimens (for detailed sample composition see Table S1), (C) lateral and (D) dorsal view of adult Falabella and (E) lateral and (F) dorsal view of the average skull shape of adult Shetland (both miniature breeds); all crania are scaled to the same length for better comparison.

Shorter rostrum

To test for rostral shortening we measured the length of and the angle between palate (LM 37–44, Table S2, Fig. S1) and basicranium (LM 49–58, Table S2, Fig. S1). The angle is expected to become smaller the larger the braincase and the shorter the palate becomes.

Smaller head to body ratio

We inspected the relationship of adult cranial length (LM 33–58) (n = 128) and average breed wither height, which we collected from the literature for a subset of 11 ponies and 18 horses (breeding guidelines for each breed, Table S3). We calculated the predicted adult cranial length of the miniature breeds as derived from linear regression of adult cranial lengths from horse breeds (Verzani, 2014) and compared it to their actual cranial lengths. The adult cranial length to wither height ratio in relation to breed is used to investigate a possible minimal limit in cranial length in the investigated breeds.

Table 1 Description of morphological differences for three age classes of medium and large breeds (0, 3, 6) and age class 6 for both miniature breeds (Falabella, Shetland) for the studied sample by module (for a detailed sample composition see Table S1).

Module	Medium and large breeds	Miniature breeds	
	Age class 0	Age class 3	Age class 6	Falabella (age class 6)	Shetland (age class 6)	
Anterior-oral-nasal	Short, very round, and narrow premaxillare; no incisors; maxillare in diastema much narrower as premaxillare; nasale and diastema are straight; diastema is very short	Elongated and broader; Third pair of incisors erupted; elongated diastema; nasale is straight or curved depending on breed; maxillare in diastema almost as broad as premaxillare	Elongated and broader; premaxilla-maxilla suture closed; elongated diastema; nasale is straight or curved depending on breed; maxillare in diastema almost as broad as premaxillare	Short, round, and broad premaxillare; maxillare in diastema almost as broad as premaxillare; all incisors fully erupted; nasale is concave	Elongated and broader; premaxilla-maxilla suture closed; elongated diastema; nasale is slightly convex; maxillare in diastema almost as broad as premaxillare	
Orbital	Round or egg-shaped depending on individual; large compared to skull length; post-orbital margin is thin	Round or egg-shaped depending on individual; medium compared to skull length; postorbital margin has thickened	Round or egg-shaped depending on individual; small compared to skull length; postorbital margin is thick	Round or egg-shaped depending on individual; medium compared to skull length; postorbital margin is thick	Round or egg-shaped depending on individual; small compared to skull length; postorbital margin is thickened	
Zygomatic-pterygoid	Frontal-zygomatic and temporal-zygomatic suture open; facial crest, zygomatic, and temporal form a straight line in lateral view	Frontal-zygomatic and temporal-zygomatic suture started to close; facial crest, zygomatic, and temporal form a straight line in lateral view	Frontal-zygomatic and temporal-zygomatic suture closed; facial crest and zygomatic form a straight line in lateral view; temporal is curved from lateral view	Frontal-zygomatic and temporal-zygomatic suture started closing; facial crest, zygomatic, and temporal form a curved line in lateral view	Frontal-zygomatic and temporal-zygomatic suture closed; facial crest and zygomatic form a straight line in lateral view; temporal is slightly curved from lateral view	
Cranial base	Round; occipital condyle and paracondylar process have a similar length; basisphenoid-presphenoid and basisphenoid-occipital suture open; basillar part of the occipital is broad	Elongated and distinct; basisphenoid-presphenoid started to close and basisphenoid-occipital suture open; basillar part of the occipital is elongated; paracondylar process is slightly longer than paracondylar process	Elongated and very distinct; basisphenoid-presphenoid and basisphenoid-occipital suture closed; basillar part of the occipital is elongated; paracondylar process is much longer than paracondylar process	Short and broad; basisphenoid-presphenoid and basisphenoid-occipital suture closed; basillar part of the occipital is broad; paracondylar process is longer than paracondylar process	Elongated and very distinct; basisphenoid-presphenoid and basisphenoid-occipital suture closed; basillar part of the occipital is elongated; paracondylar process is much longer than paracondylar process	
Cranial vault	Occipital is not fused to any other bone; very round; frontal-parietal-occipital doming; occipital crest very small	Frontal-parietal doming; occipital elongated; occipital crest more pronounced; occipital started fusing to surrounding bones	Frontal-parietal-occipital flattened; occipital elongated; occipital crest very pronounced; occipital mostly fused with surrounding bones	Frontal-parietal-occipital doming; occipital elongated; occipital crest very pronounced; occipital mostly fused with surrounding bones	Frontal-parietal-occipital doming; occipital elongated; occipital crest very pronounced; occipital mostly fused with surrounding bones	
Age classification	First post-natal stage, before the eruption of the first pair of incisors, up to 1 week old	Time after the eruption of the third pair of incisors until the eruption of the first molar, six month to one year, before sexual maturity; weaning is around 6 month of age	Last age stage after the eruption of the third molar, from 4 years on, skeletal maturity	Adult, age stage 6	Adult, age stage 6	

Results

Characterization of cranial shape of miniature and horse breeds

We calculated and visualized the mean shape for each age class from Procrustes shape data, as well as that of the adult stage of the two miniature breeds (Fig. 2). A description of the different age classes (0, 3, 6) and the adult crania of the Falabella and Shetland pony are presented in Table 1. The juvenile age classes of horses are characterized by a very broad and short cranium with a bulging anterior-dorsal part of the braincase (Fig. 2). During growth, the cranium elongates (rostrum stronger than the anterior part of the braincase) and the anterior-dorsal part of the cranium flattens. The orbit size decreases in relation to the complete cranium.

Cranial ontogenetic shape change of pony and horse breeds

PCA (Fig. 3A) reveals that the ontogenetic stages 0–6 separate in PC1–PC2 space along PC1 from adult (PC1 negative) to juvenile age stages (PC1 positive). The first PC accounts for 47.1% of the total shape variation. A gap in the shape space between age classes 0–2 and 3–6 is visible. Also, we can observe that cranial shape among early ontogenetic stages is more similar to each other than cranial shape of the adult stage, where the scatter becomes larger. When comparing the miniature breeds to horses in shape space, it becomes evident that the Shetland specimens align with the respective age classes of larger breeds, but constitute the most ‘youthful’ cohort of the respective stages (Fig. 3A). The adult skull (age class 6) of the smallest of all horse breeds, the Falabella specimen, clusters most within age stages 3 and 4 of regular-sized breeds (Figs. 3A and 4). The same PCA highlighted by group—miniature breeds, ponies, and horses—(Fig. 3B) visualizes that horse breeds scatter more along PC2. The contrast of shape change along PC 1 (Fig. 3C) demonstrates that the braincase undergoes larger changes than the rostrum.

Figure 3 Principal component analysis of 194 specimens of 25 horse and 13 pony breeds, including two miniature breeds.

(A) PC1-PC2 scatterplot shows ontogenetic trajectory for all analysed breeds (see Table S1 for details). Miniature breeds, breeds of extreme small size are highlighted with diamond and star shapes. Colours represent age classes. (B) PC1–PC2 scatterplot as (A) but the three groups—horses (filled black circles), ponies (filled grey diamonds) and miniature breeds (open triangle for Shetland and star for Falabella) are highlighted. (C) Dorsal and (D) lateral views of the cranium show the shape changes along PC1, adult shape in grey and juvenile shape in black. TPS-grids can be found in Fig. S3.

Analysis of ontogenetic allometry within and between pony and horse breeds

We tested for allometry and differences in allometric growth between ponies and horses performing Procrustes ANOVAs (analyses of variance). Regression of skull shape (Procrustes coordinates) on log centroid size for the entire sample (ponies and horses) shows a strong effect of size on shape (R2 = 0.42, p = 0.001). Adding “type” as an additional covariate yielded that mean shapes of ponies and horses (F = 9.201, Z = 6.488, p = 0.001) as well as their allometries (F = 3.613, Z = 4.141, p = 0.001) differ. Specifically, they differ in rate of change (Z = 4.125, p = 0.001) and slope (angle = 21.25 deg, r = 0.93, Z = 5.28, p = 0.001). This result can be visually assessed in Fig. 4B. A test of least square (LS) means revealed no difference in intercept (Z =  − 1.089, p = 0.856). Generally, ponies have attained more maturity than horses at the same size, while exhibiting the same skull shape as horses. The position of the adult Falabella (age class 6) skull in the shape-size regression (Fig. 4A) and allometric trend (Fig. 4B) indicate that the Falabella skull shape resembles younger horse breed shapes at the same size.

Figure 4 Allometric trend in pony and horse skull shapes.

(A) Regression of shape on log centroid size and (B) predicted shape values from regression of shape on log centroid size. Ponies are shown in stars, horses in open squares. The position of adult Falabella skull is highlighted. Colours correspond to age classes.

Testing features of “paedomorphosis” in miniature breeds

During ontogenetic growth of horses we observe that orbits grow smaller in relation to cranial length (Fig. 5A), and the basicranium becomes shorter relative to the length of the rostrum (Fig. 5B). The angle between the basicranium and palate does not differ significantly among the different age stages (Fig. 5C). The growth pattern of Shetland ponies is similar to that of horses and do not show any signs of enlarged orbits or shortened rostra or increased brain case in the adult stage (Figs. 5A–5C). The adult Falabella exhibits larger orbits (Fig. 5A) but otherwise no other juvenile features regarding the rostrum or braincase (Figs. 5A–5C). The predicted cranial length of the Falabella and Shetland derived from linear regression of adult cranial lengths from horse breeds is 24.6 cm and 31.1 cm respectively, which contrast to their actual lengths of 35.6 cm and about 39 cm, respectively. Both actual cranial lengths fall slightly outside of the 95% prediction interval (Fig. 5D). The cranial length at the upper prediction limit raises the question whether ponies are constrained to have larger crania. Examination of the cranial length to wither height ratio in relation to breed ordered by increasing maximal wither height supports that the adult miniature breeds, Shetland and Falabella, have larger crania relative to wither height than their horse cognates (Fig. 5E).

Figure 5 Testing for “paedomorphic” features.

(A–C) Orbit to cranial length ratio, basciranium to palate length ratio, and angle between basicranium and palate, per age category, all breeds except for miniature breeds as boxplots in grey, and Shetland (diamonds) and Falabella (star) as miniature breeds superimposed in black. (D) Adult cranial length in relation to maximal wither height with 95% confidence interval (solid line) and prediction interval (dotted line), and regression line (red). Regression line and 95% prediction curves were extended to smaller wither heights to accommodate the cranial length of the miniature breeds in the same plot. (E) Adult cranial length to wither height ratios were plotted against breed in ascending order from the smallest breed, the Falabella, to the largest breed, the Shire horse. Abbreviations for the breeds: fab, Falabella; she, Shetland; exm, Exmoor pony; wel, Welsch, mon, Mongolian; kon, Konik; bos, Bosnian pony; huz, Huzule; scp, Scottish pony; ice, Icelandic Horse; hny, Hackney; arb, Arab; grp, German Riding pony; grb, Grisons; lpz, Lipizzan; piz, Pinzgau; nor, Norik; ano, Anglo-Norman; thb, English Thoroughbred; hun, Hungarian; trk, Trakehner; han, Hannoverian, odb, Oldenburgian; suf, Suffolk; kdr, Kladrubian; blg, Belgian Draft; hol, Holstein; cds, Clydesdale; shi, Shire; for the ten breeds that were included in 3D GM but not linear analyses, no information on average wither height could be found.

Discussion

Horses show allometric cranial growth where the juvenile specimens are significantly different in cranial shape from the adult specimens (Sánchez-Villagra et al., 2017) and this study (42%), as has been attested for other domesticated species (Wilson, 2018) and mammals in general (Porto et al., 2009), The largest shape differences in PC1–PC2 shape space in our sample can be found between the age classes 0–1 and 2–6. The difference between those two age clusters is most likely caused by the low sample size in age class 2 (n = 2) and it is likely that the ontogenetic trajectory for horses and ponies would form a continuum if age class 2 would contain more specimens. To our knowledge, based on our examination of many museum collections, specimens of that age class are rarely available.

Ponies and horses show differences in ontogenetic trajectory direction or angle but not in intercept, which indicates that juveniles of all breeds start with a similar skull shape but diverge during ontogenesis. Furthermore, ponies (including miniature breeds) develop faster than regular-sized breeds and stop growth earlier. Therefore, horses develop adult cranial shapes that ponies do not reach during growth. As a result, a younger horse specimen and an older pony specimen can exhibit the same skull shape. This pattern could explain why ponies look like juvenilized adult horses. Only the miniature breed Falabella, but not the Shetland, exhibits a “paedomorphic” feature: enlarged orbits relative to cranial length in its final age state.

Given our results, we would like to emphasize that there are likely inter-breed differences in skull shape at birth and possibly prenatally but shape data for those age classes are scarce. We have preliminary evidence that skull shape differences among pony breeds arise prenatally. We compared the ontogenetic trajectories of the Shetland as a miniature breed and the Welsh as a regular pony (Fig. S2, not presented in main text due to small sample size). Apart from dogs (Werneburg & Geiger, 2017), this pattern has also been hypothesized in pigs (Evin et al., 2017), and needs to be further investigated in horses including more breeds.

We investigated whether ponies, when compared to horses, represent a case of craniofacial evolutionary allometry (CREA) (Cardini & Polly, 2013; Cardini et al., 2015; Tamagnini, Meloro & Cardini, 2017). CREA predicts that larger forms are long-faced and smaller forms short-faced as a sign of paedomorphism. As an approximation for braincase-to-face relationship we calculated the ratio of basicranium-to-palate lengths for miniature and horse breeds. We found no signs of CREA. In contrast, we found that our results are in accordance with the long-face hypothesis for grazing ungulates (Spencer, 1995). The long-face hypothesis does not offer a definite explanation why longer faces are observed in smaller forms of grazing ungulates. In the case of the miniaturized horses, we propose that this could be due to constraints in tooth morphology and the feeding style of grazing. Among veterinarians, it is commonly known that miniature horse breeds have a higher requirement for veterinary dentist procedures, due to their almost regular horse-sized teeth (Wilson, 2012). The same health-related problem has been shown for pet rabbits, which experience a rostral shortening through domestication without a change in dentition (Böhmer & Böhmer, 2017). These functional constraints have been investigated also in humans showing that miniature forms tend to have relatively larger teeth than regular-sized forms (Shea & Gomez, 1988). Since horses and ponies feed on a very nutrient poor diet, they are in need of a highly specialized feeding apparatus to ensure the best energy recovery possible. A strong shortening of the rostrum, as can be found in some dog and pig breeds (Geiger & Haussman, 2016; Evin et al., 2017), is most likely possible due to their energy rich diet (carnivore/omnivore) that can be exploited with fewer, smaller, or differently placed teeth. In cows, which also feed on nutrient poor grass, one case of rostral shortening is known: the Niata breed (Veitschegger et al., 2018); this was likely possible due to the more efficient uptake of nutrients through rumination. There is evidence of rostral shortening in a extinct herbivorous, non-ruminant browsing mammal clade, the short-faced kangaroo subfamily Sthenurinae (Prideaux, 2004). Yet to our knowledge, there is no evidence of the existence of a grazing ungulate with rostral shortening.

Large heads and increased head-to-body ratio has been shown to be a paedomorphic feature in other domesticated species, namely dogs and chicken (Alberch et al., 1979; Gould, 1980; Wayne, 2001), but does not associate with paedomorphosis in horses as new born foals have shorter rostra than adult individuals because teeth development drives rostral lengthening; as well as foals having relatively longer legs (Habermehl, 1975; Van Heel et al., 2006; Goodwin, Levine & McGreevy, 2008) (Table 1). The long legs in horses are a necessity for survival, since new born foals are very precocial and need to keep up with the herd from day one. Our study did not compare the actual limb to head length ratio which has been proposed as a sign for paedomorphism (Goodwin, Levine & McGreevy, 2007; Goodwin, Levine & McGreevy, 2008), but used wither height as a proxy for size. We found that the miniature breeds have larger heads relative to body size (approximated by wither height) when compared to horse breeds. So adult miniature breeds do not exhibit the juvenile state of a horse in respect to head-to-body size ratio.

Regarding the adult Falabella skull and the case of the Falabella in general, we could not support our first subjective impression that the Falabella must be a “paedomorphic” horse. However, this impression probably derived from the very round anterior-dorsal part of the braincase, whose geometric morphometric quantification by using true landmarks eluded us due to the lack of sutures in that portion of the cranium, but the curvature can be seen in the photograph of the Falabella cranium (Fig. 1A). Additionally, the Falabella does exhibit a less downward curved rostrum than adults of regular-sized breeds (Fig. 1). For a better assessment of a rounded cranium roof, future investigations are advised to use semi-landmarks or polygons (MacLeod, 2013; Collyer, Sekora & Adams, 2015).

Conclusions

We investigated patterns of allometry during ontogeny in horses as a case of directional artificial selection for extreme size differences in domesticated horses. Wither heights range from 84 cm in the Falabella to 178 cm in the Shire horse. We looked at allometric trends between ponies and horses and investigated typical patterns of “paedomorphosis”, defined as juvenile appearance in adult stage, with an emphasis on miniature horses as an extreme case of size selection. We found that ponies and horses have shifted ontogenetic trajectories that vary in length but not in direction or angle, with the consequence that ponies exhibit similar skull shapes at older age stages as horses at younger age stages. This pattern is a potential source of the perceived juvenilized skull shape of ponies, additionally to features of the postcranial architecture, namely general small size or short limbs, or behavioral aspects as has been show in dogs (Hare & Woods, 2013), such as the use of body language (Goodwin, Bradshaw & Wickens, 1997) or facial expressions (Waller et al., 2013).

Other than the overall shape development of ponies halting earlier than of horses we find no other evidence of “paedomorphic” features, as enlarged orbits, shorter faces, or increased head-to-body ratio, as it is the case in dogs and pigs, except for the adult Falabella skull. Miniature breeds (Falabella and Shetland) have increased skull-to-body ratio when compared to horse breeds. This could be due to the very essence of horses and ponies, that is being a grazing ungulate as postulated in the long-face hypothesis (Spencer, 1995). We propose functional and metabolic constraints rather than flight responses as a potential driver of this pattern.

Supplemental Information

Table S1 Raw data: classifiers and landmark coordinates

Not to be read into R.

Click here for additional data file.

Data S1 R code used for the analysis of ontogenetic allometry in horse skull shape data

Click here for additional data file.

Data S2 Input file for analyses in R, containing 3D geometric morphometric landmark data for pony and regular sized horse breeds, as well as classifiers describing the individual data points, i.e.specimens

Use this file to run the commands in the rmarkdown file.

Click here for additional data file.

Data S3 Input data file necessary for indicating landmarks of matching symmetry during Procrustes superimposition

Click here for additional data file.

Data S4 Wireframe used for plotting purposes during 3D GM analyses of horse skulls

Click here for additional data file.

Data S4 R code for the analysis of linear measurements of horse skull features that were used to create Figure 5

Click here for additional data file.

Supplemental Information 1 Supplemental Figures and Tables

This document contains:

Table S2: Description of the landmarks, including position, type, and to which module it belongs, collected on each cranium.

Table S3: Average wither height in cm for each breed, with reference.

Figure S1: Depiction of landmarks used in this study.

Figure S2: Principal component analysis of ontogenetic series of Shetland pony and Welsh specimens.

Figure S3. Ontogenetic skull shape changes from juvenile to adult stage in thin-plate-spline representation.

Click here for additional data file.

We thank the many institutions and people giving us access to their collections: Christiane Funk and Frieder Mayer (MfN Berlin, Germany), Renate Lücht (Institut für Haustierkunde, Christian-Albrechts-Universität of Kiel, Germany), Renate Schafberg (Museum für Haustierkunde “Julius Kühn”, University of Halle, Germany), Frank Zachos, Alexander Bibl, Konstantina Saliari and Erich Pucher (NHW Vienna, Austria), and Alfredo Carlini (MLP La Plata, Argentina). We thank Laura A.B. Wilson, Madeleine Geiger for comments on an earlier version of the manuscript, and Emma Sherratt and Eric Scott for their helpful comments during the review process.

Additional Information and Declarations

Competing Interests

Author Contributions

Data Availability

The authors declare there are no competing interests.

Laura Heck conceived and designed the experiments, performed the experiments, analyzed the data, prepared figures and/or tables, authored or reviewed drafts of the paper, approved the final draft.

Marcelo R. Sanchez-Villagra conceived and designed the experiments, contributed reagents/materials/analysis tools, approved the final draft.

Madlen Stange conceived and designed the experiments, analyzed the data, prepared figures and/or tables, authored or reviewed drafts of the paper, approved the final draft.

The following information was supplied regarding data availability:

The raw data (including 3D landmark coordinates and classifiers for all specimens measured) and the codes for conducting the analyses are available as Supplemental Files.

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
