# Peer review of "Why the long face? Comparative shape analysis of miniature, pony, and other horse skulls reveals changes in ontogenetic growth"

_PeerJ, doi:10.7717/peerj.7678_

## Round 0.1 · original submission · Major Revisions

Congratulations, both reviewers found your manuscript to be a valuable contribution and both recommend publication, presuming that the issues they have identified are satisfactorily addressed.

I've carefully read all of the reviewers' comments and they seem apt to me. Because this manuscript stands or falls on the statistical work, please pay special attention to the detailed feedback of Reviewer 1. In particular, Reviewer 1 notes, "The R code provided with the text does not show the analyses for the coordinate data provided so I cannot check what was done." Including that information in the revised manuscript should alleviate the concern, and it will make the paper more useful to readers down the line.

Both reviewers also drew attention to the uses of "adult", "regular-sized", "miniature", and "pony" as points of potential confusion - probably best to define all of these terms explicitly in the Introduction or Materials & Methods and be rigorous about using them only as defined.

Please take all of these comments in the constructive spirit in which they are intended. I will look forward to seeing an improved version of this work in the near future.

·

Basic reporting

Overall the manuscript is written well, clear and concise. The literature cited is appropriate and the hypotheses are clearly stated.

My only concern on the reporting is that the introduction does not follow the usual format of broad to narrow and as such is difficult to follow. Suggest some rearrangement, and move the final horse paragraph up to after the first paragraph, since the breed sizes should be introduced before they are discussed in terms of hypotheses of allometry.

I would like to see some clarifications in the manuscript regarding horses, ponies and miniature ponies, because the terms are not always referred to consistently (and I believe the authors use pony to refer to specifically the miniature breeds at times, e.g. Line 326). The authors mention in the introduction that ponies are defined as horses of a particular height, and miniature ponies are a height below this. In the caption of Figure 1, they use a “Welsh” pony as an example of a “regular sized horse”, which is confusing since it is not clear if ponies and horses are distinct in morphology or just size? Furthermore, on lines 273-278, the authors discuss ponies and regular-sized horses in relation to the CREA hypothesis. Finally, caption for fig.4 simply states small and large breeds. Please clarify and make consistent.

Experimental design

The authors amassed a very commendable dataset of skulls from museums for their study. The methods applied to test for allometry and consider paedomorphosis are appropriate.

Pooling the data for all non-miniature breeds of horses and ponies assumes a single allometric slope for these breeds. As such, I suggest below that the authors plot the regression residuals and not simply the regression predicted values, to show whether this is a valid assumption.

Only some R functions mentioned explicitly in the methods section, please be consistent.

Rstudio is simply a graphical user interface (GUI) for R and doesn’t really need mentioning.

The outlier test implemented in geomorph is simply a data inspection tool and is not necessarily identifying statistical outliers, but simply specimens than fall far from the average shape. In this case, it is clear the juvenile forms are more different, but they are not actual statistical outliers. Omit figure S2 and discussion of this from manuscript.

Validity of the findings

I have a concern about the shape data, which I’m alerted to based upon two lines of evidence. First, on line 224 the reported R-squared value is extremely high; higher than I’ve seen for shape allometry studies (which are normally lower due to the high dimensionality of the data). Secondly, the values on the y-axis of figure 4 are orders of magnitude higher than normally seen for predicted values of Procrustes residuals (I’d expect 0.2 rather than 2000). The R code provided with the text does not show the analyses for the coordinate data provided so I cannot check what was done. But seeing these values I’m concerned the Procrustes residuals are not calculated correctly or are not being used in this analysis? These values seem like they are in the original pre-GPA coordinates. Please check and confirm.

The authors should give versions for the R packages used. I am not sure what version of geomorph was used, but I am aware of a known bug in version 3.0.7 on CRAN regarding the procD.allometry HOS test (and was fixed on the version in GitHub). The bug was reading from the wrong column of the ANOVA table when it prints out whether to reject or support the null hypothesis. It is worth double-checking this if the authors were using v3.0.7 from CRAN.

Figure 4 – It would be helpful to the reader to see a ‘regression score’ (sensu Drake & Klingenberg 2010) version of this plot in order to see the regression residuals and thus variation in the data (to be able to confirm that the r-squared value of 0.94 is correct), in addition to the predicted lines.

Additional comments

This is an interesting paper and important for our understanding of how allometry contributes to morphological diversity.

Line 296-297 Perhaps I misunderstand wat the authors are saying here, but what about the extinct short-faced kangaroo subfamily Sthenurinae?

Figure 1 – Inconsistency in naming. Call all ‘XX pony’, or just give name without pony suffix.

Figure 3A – This is a wonderful graph! Can you make it bigger?

Figure 3B – I think that thin-plate spline (TPS) grids would work here to show the overall shape change from juve to adult, rather than the two configurations overlaid.

Following my comments above on horses, ponies and miniature ponies, it would be good to see another graph such as a PCA like fig. 3 showing miniature ponies, ponies and horses as three separate groups/trajectories, since there is not much dialogue about whether ponies are distinct from horses, or is it just miniature ponies that are distinct?

·

Basic reporting

A few minor grammatical issues are noted in the appended pdf.

Experimental design

No comment

Validity of the findings

No comment

Additional comments

In general, I thought the paper was well written, well presented, and addressed an issue of some interest. The findings were not what I would have anticipated, and I'll be glad to see these data and interpretations available.

---

## Round 0.2 · Minor Revisions

Congratulations, the reviewer found your manuscript improved and only a few small things now need to be addressed before it can be accepted for publication. Please be diligent in addressing the reviewer's remaining concerns, either in the manuscript or the rebuttal letter. I agree with the reviewer that the title could be improved, both to better represent the content of the paper, and to hopefully get the work the attention that it deserves.

·

Basic reporting

Clear and unambiguous. Only minor concern: “Regular sized horses” – Since regular-sized horses (requires hyphen) are defined with a wither height greater than that of ponies, it would be clearer to just call them horses throughout after definition. Else it suggests there are horses of a size other than that which is defined. Regular-sized breeds compared to miniature breeds would be valid terminology. But then that would confuse whether you are talking about ponies or horses.

Experimental design

Methods described with sufficient detail & information to replicate; all code and data are presented in the supplementary materials.

Validity of the findings

The findings are novel and supported by the data. My only concern is that given the size-range sampling for horses is not as good as for ponies (with only 3 small specimens). As such, it may be that the observed allometric slope difference, which is relatively small, is artefactual due to the sampling. This caveat should be given in the text.

Additional comments

I have reviewed the revised manuscript. I am glad to see that the allometric analyses was a simple error that could be easily rectified. 42% of the variation is very reasonable. I apologise for missing the R markdown file in the original submission; this is done well and I have no comments to make. The changes made are great, particular the new graphs. I have only some small further comments on the manuscript, provided in each section and below:

The title doesn’t do the paper justice since there is no mention of the allometric hypotheses. Suggest changing to include the key words of allometry and miniature.

Lines 446-448 – Arguably all vertebrates show allometric growth in crania (as well as other body parts), not just domesticated animals. Perhaps instead here make the observation about generally common allometric changes among all of the breeds examined, as shown by the 42%.

---

## Round 0.3 · accepted · Accept

Thank you for your diligence in addressing the review comments. I am happy to accept your manuscript for publication in PeerJ.

The decision of whether or not to publish the peer reviews alongside the paper is entirely yours, and will not affect how your paper is handled going forward. However, I encourage you to do so. Making the reviews public allows the reviewers to receive credit for their efforts, and also contributes to the emerging culture of fairness and transparency in editing and peer review.